# 'Experiences of patients and their informal caregivers with cognitive stimulation programs for dementia: A qualitative systematic review protocol'

**Simone M. Ryan** [1]*, **Manigandan Chockalingam** [1], **Orla Brady** [1,2]

1 Department of Occupational Therapy, School of Health Sciences, University of Galway, Galway, Ireland,
2 Trim Primary Care Centre, Knightsbridge Village, Trim, Co. Meath, Ireland

* s.ryan72@nuigalway.ie

## Abstract

### Introduction

Cognitive stimulation, an individual or group intervention approach aiming to improve cognitive and social functioning among individuals with mild-to-moderate dementia, is often considered a complex intervention. The patient's experience of a complex intervention is unique and often determines its effectiveness. This proposed qualitative systematic review aims to comprehensively synthesise the experiences of individuals with dementia and their informal caregivers who have participated in cognitive stimulation programs, identifying perceived benefits, challenges, barriers, and facilitators to this approach to intervention.

### Methods

This review will consider qualitative studies that evaluate the experiences of individuals with a diagnosis of dementia and/or the informal caregivers of individuals with dementia who have participated in a cognitive stimulation program. Searches will be conducted across MEDLINE (Ovid), Embase (Elsevier), PsycINFO, Scopus, CINAHL (EBSCO) and Web of Science. Quality of eligible studies will be assessed using the JBI Critical Appraisal Checklist for Qualitative Research, and a standardised data extraction tool in JBI SUMARI will be used to extract data from relevant studies. The meta-aggregation approach will be used to pool qualitative research findings, which will then be synthesised to produce a single set of findings in narrative format.

### Discussion

This qualitative systematic review will identify and synthesise the evidence regarding the experiences of individuals with dementia who have taken part in a cognitive stimulation program and the experience of their informal caregivers. As a variety of cognitive stimulation programs exist, our findings will summarise the experiences of these interventions to inform the future development and delivery of cognitive stimulation programs.

**Data Availability Statement:** No datasets were generated or analysed during the protocol. All

relevant data from this study will be made available upon study completion.

**Funding:** The authors received no specific funding for the completion of this work.

**Competing interests:** The authors have declared that no competing interests exist.

## Trial registration

**PROSPERO registration number:** CRD42022383658.

## 1. Introduction

Dementia is a clinical syndrome characterised by a progressive decline in one or more cognitive domains, adversely impacting an individual's social and functional performance [1, 2]. Dementia can also be referred to by major neurocognitive disorder, as per the updated terminology of the Diagnostic Statistical Manual of Mental Disorders, 5th edition, (DSM-5) [1]. Several studies have linked the disorder's cognitive decline to decreased quality of life, increased caregiver burden, and higher care costs, contributing to dementia being one of the leading causes of hospitalisations and admissions to skilled nursing facilities [3, 4]. Globally, approximately 57.4 million people are living with dementia, which is predicted to rise to 152.8 million by 2050. This estimated increase in prevalence of the disease is reflected within the Irish population, as numbers of people living with the condition are expected to rise from approximately 53,932 in 2019 to 142,416 by 2050 [5]. Consequently, the societal and monetary costs of dementia are anticipated to rise in conjunction with the prevalence of the disease. Hence, effective, and feasible interventions are necessary to slow the cognitive and functional decline of those with dementia [2, 5]. Several non-pharmacological interventions are currently being used to address these declines, with cognitive stimulation being the most frequently used intervention with individuals with mild-to-moderate dementia [6].

Cognitive stimulation is a broad term encompassing a variety of psychosocial, cognition-focused interventions widely used in dementia care. Cognitive stimulation interventions are typically provided in a stimulating and rewarding group setting, where participants engage in various activities and discussions to improve their overall cognitive and social performance, as opposed to a focus on specific functions in isolation [7, 8]. There is a wide range of cognitive stimulation activities described in the literature, including discussions of past and present events or topics of interest, word games, puzzles, music, or practical activities like gardening or baking [9]. Similarly, many cognitive stimulation approaches have been described in the literature, including manualised therapies, individual programs, and cognitive stimulation delivered as part of a wider multi-component intervention [10]. One notable example of a manualised cognitive stimulation intervention is Cognitive Stimulation Therapy (CST). CST, developed by Spector et al. [11], is a commonly used, group intervention based on cognitive stimulation principles. CST sessions take place twice weekly for seven weeks, with each week's activities focused on a particular theme. Several modifications to this CST program have been made in response to specific clinical needs, including a 24-week maintenance program (MCST) and an individualised program (iCST). While CST and MCST have demonstrated significant improvements in cognitive performance, communication, and quality of life among individuals with mild-to-moderate dementia [9, 11], these benefits have not been replicated with iCST [12], suggesting that group cognitive stimulation programs are more effective than individual programs.

Complex interventions, such as cognitive stimulation, have several interrelated characteristics, including the number of components, the range of behaviours targeted, and the level of expertise required of those who deliver such interventions, in addition to the complexity inherent in group interventions [13]. The combination of all these complexities often makes the experience of undergoing complex interventions particularly valuable, meaningful and unique. A complex intervention's experiences are often considered an important factor in determining

its effectiveness, which prompted the Medical Research Council to advocate qualitative evaluation as a recommended practice [14], and cognitive stimulation is no exception. Several qualitative studies have examined the experiences of individuals with dementia who participate in CST, and they have reported various perceived benefits and challenges. For CST and other cognitive stimulation programs to be improved and implemented effectively, systematic documentation and synthesis of these findings are essential.

The majority of people with dementia live at home in their community [15]. However, due to the cognitive and social changes characteristic of the condition, they typically require assistance to continue engaging in meaningful activities of daily living [15]. It is estimated that 75% of this at-home care is provided informally, without pay, by family, spouses, or friends [16]. However, the consequences of dementia affect not only those who live with the condition but also those who provide care for them. While informal caregivers may be motivated to provide care due to a sense of duty, fulfilment or reciprocity, long-term caregiving for individuals with dementia has been shown to lead to a variety of negative personal consequences, including burnout syndrome [15]. Burnout syndrome is characterised by emotional exhaustion and loss of personal fulfilment and has been found to interfere with the quality of patient care, potentially resulting in early institutionalisation [17]. As dementia care and management rely significantly on the support of informal caregivers, it is therefore critical to consider the informal caregiver's experiences in all aspects of dementia management, including their perspectives on cognitive stimulation interventions. The findings of previous qualitative studies have provided valuable insights into the experiences of caregiving as well as the impact of cognitive stimulation interventions on the relationship between informal caregivers and individuals with dementia [18]. A systematic review of the impact of CST groups on the caregiver's experience of everyday life conducted by Lauritzen et al. [18] reported caregivers experienced feelings of enrichment and happiness due to improvements in their care recipients' communication and behaviour after participating in a CST program [18]. Furthermore, caregiver perspectives are valuable to the evaluation process of complex interventions since conducting research solely with the person with dementia can restrict results due to the memory challenges associated with the condition [19]. Thus, the caregivers' experiences of cognitive stimulation programs are as equally important as that of the experiences of individuals with dementia.

To conclude, it is necessary to systematically synthesise the experiences of individuals with dementia who participate in cognitive stimulation programs and their informal caregivers. A preliminary search for existing systematic reviews on this topic was conducted on PROSPERO, JBI Evidence Synthesis, the Cochrane Database of Systematic Reviews and PubMed. Two similar qualitative systematic reviews were identified: those of Gibbor et al. [20] and Tuomikoski et al. [21]. The scope of Tuomikoski et al.'s review, which examined the experiences of people with progressive memory disorders who have participated in non-pharmacological interventions, was broader than the proposed review since it evaluated cognitive stimulation alongside other non-pharmacological interventions, such as exercise programs and music therapy [21]. Gibbor et al., on the other hand, focused their review on a specific type of cognitive stimulation intervention: CST and its variations, excluding other forms of cognitive stimulation interventions [20]. While both these reviews are rigorous, of high quality, and critical in synthesising the experiences of their respective interventions of interest, neither review gives a comprehensive summary of the experience of participating in all types of cognitive stimulation interventions. The inclusion of all types of cognitive stimulation intervention programs, such as non-manualised, individual, or multi-component programs, as well as CST, would provide a more comprehensive meta-synthesis of the experience of participating in a cognitive stimulation intervention. This summary would be valuable in the future development and implementation of interventions based on cognitive stimulation principles for individuals with dementia.

The objective of this review is to identify and synthesise the evidence regarding the experiences of individuals with dementia and their informal caregivers who have participated in all types of cognitive stimulation programs, identifying the perceived benefits, challenges, barriers, and facilitators to this approach to intervention.

## 2. Methods

### Review question

What are the experiences of individuals with dementia and their informal caregivers of a cognitive stimulation program?

### Inclusion criteria

**Types of participants.** This review will consider studies that include participants with a diagnosis of dementia and/or the informal caregivers of individuals with dementia. The term dementia, in this review, refers to a clinical syndrome where an individual experiences a progressive decline in cognition that interferes with occupational, domestic, or social functioning. Only progressive forms of dementia will be considered in this review, as non-progressive dementias typically resolve with appropriate treatment, and cognitive stimulation is not designed for this population [22]. Examples of progressive forms of dementia include Alzheimer's disease, vascular dementia, dementia with Lewy bodies, frontotemporal dementia and Parkinson's disease dementia. Individuals with mixed forms of dementia, i.e., vascular dementia and Alzheimer's disease, will also be included. Studies that include participants with a mild neurocognitive disorder, and no diagnosis of dementia, will be excluded, as cognitive stimulation was also not designed for this population [8]. While cognitive stimulation, as previously discussed, is designed for use with individuals with mild-to-moderate dementia, studies that include participants with severe dementia will therefore be excluded. However, there is the possibility that studies may not declare the level of cognitive impairment of their participants with dementia. It would be considered in this case that the participants in these studies had mild-to-moderate dementia and will be included. Furthermore, studies that evaluate the experience of individuals with dual diagnosis, i.e., individuals with dementia and intellectual disability, will be excluded, as this population require an adapted approach to cognitive stimulation due to the pre-existing cognitive deficits and sensory impairment [23, 24].

In this review, studies investigating informal caregivers of individuals with dementia will also be considered. The term informal caregiver refers to an individual who provides unpaid assistance with everyday tasks, including but not limited to physical, emotional, functional, and household tasks, regularly. Informal caregivers are often close family members, spouses, or friends who provide unpaid care to individuals with dementia. Informal caregivers will be considered regardless of their age, gender, cultural background, or relation to the individual with dementia [25]. Those who receive pay for their services as formal caregivers will be excluded from this review, irrespective of whether they have had training or education in providing care.

**Phenomena of interest.** The phenomenon of interest in this review is the experience of individuals with dementia and the experience of their informal caregivers who have participated in a cognitive stimulation intervention program. Studies must identify that they are evaluating the experiences of individuals with dementia and/or their informal caregivers of a cognitive stimulation program or an intervention based on cognitive stimulation principles. Cognitive stimulation interventions involve the engagement of participants in a range of activities and discussions that aim to enhance global cognitive and social functioning [8]. Activities can range from discussions of past and present events, topics of interest, word games, puzzles,

music, or practical activities like baking [9]. All types of approaches will be considered, including reality orientation programs, individual or group programs, virtually delivered programs, manualised programs like CST and its adapted formats, or the delivery of cognitive stimulation as part of a broader, multi-component intervention. The individuals' experiences can include experiences during or after undergoing cognitive stimulation interventions, and the experience may be positive, negative, or neutral. No restrictions will be placed on the number of sessions completed or the treatment duration. Experiences expressed by staff will be excluded as this is not the focus of this review.

**Context.** In this review, there will be no restrictions on the context, and cognitive stimulation in any settings (e.g., community-based, inpatient, outpatient, nursing home) will be considered for inclusion. The review will not be restricted to geographical locations.

**Types of studies.** This review will consider studies that focus on qualitative data, including but not limited to designs such as phenomenology, grounded theory, ethnography, and qualitative descriptive designs. Qualitative results of mixed method studies will also be considered.

## Methods

This systematic review will follow the JBI methodology for systematic reviews of qualitative evidence [26] and will adhere to the Preferred Reporting Items for Systematic Reviews and Meta-Analyses (PRISMA) [27]. The Preferred Reporting Items for Systematic Review and Meta-Analysis Protocols (PRISMA-P) 2015 statement [28] was used to develop the qualitative systematic review methodology (refer to S1 Checklist in Supplementary Information).

**Search strategy.** This review will consider both published and unpublished literature. The search strategy will be completed in a three-step process. Firstly, a limited search of MEDLINE and CINAHL was completed to identify studies on the topic. Text words in the titles, abstracts and index terms used of relevant studies from this search were utilised to develop a search strategy for MEDLINE (full search strategy detailed in S1 File in Supplementary Information). The search strategy will be adapted for each included database and literature source. Databases searched will include MEDLINE (Ovid), Embase (Elsevier), PsycINFO, Scopus, CINAHL (EBSCO) and Web of Science. Grey literature will be searched for through Google Scholar (first 100 hits only) and PsyArXiv. In addition, articles published since 2010 in *Alzheimer's and Dementia Journal*, *Aging & Mental Health* and *Dementia* will be hand-searched. The final step in this search strategy will involve screening reference lists of included studies. Apart from the hand-searched journals, every other database search will have no restrictions on the date of publication. As English is the primary language of the reviewers with limited access to translation facilities, only articles published in English will be considered for inclusion. While this limits the scope of the review, there is a significant threat that the translation of non-English studies may lead to misinterpretation or the loss of meaning of certain phenomena [29], and hence non-English studies will not be considered for inclusion. However, to ensure transparency in the search process, the search will be conducted inclusively with no language limits, and non-English studies will be excluded at the time of screening.

**Selection process.** All citations following the search will be collated, exported to Rayyan, and deduplicated. The eligibility criteria will be tested on a sample of citations (between 6 and 8 articles, including those determined to be definitely eligible, definitely ineligible, and doubtful) to ensure consistency in the application of the criteria [30]. To determine whether the titles and abstracts satisfy the review's inclusion criteria, two independent reviewers will screen them. Studies that are deemed to be potentially relevant will be retrieved in full and will be imported into the JBI System for the Unified Management, Assessment and Review

of Information (JBI SUMARI) [31]. The full text of all selected citations will be assessed by two independent reviewers against the inclusion criteria, and the reasons for excluding full-text papers that do not meet the criteria will be recorded and reported in the systematic review. When disagreements arise between the reviewers at each stage of the selection process, they will be resolved through discussion or by involving an additional reviewer. In the final systematic review, the results of the search and the selection of studies will be presented in accordance with the Preferred Reporting Items for Systematic Reviews and Meta-Analyses (PRISMA) guidelines [27].

**Assessment of methodological quality.**   A critical appraisal of the methodological quality of eligible studies will be conducted by two independent reviewers using the standard JBI critical appraisal checklist for qualitative research [32]. As needed, the authors of papers will be contacted for clarification and to request missing or additional data. Disagreements between the reviewers will be resolved through discussion or by involving an additional reviewer. Critical appraisal results will be presented in a narrative format and in table format. To ensure that all experiences are captured comprehensively and no evidence is missed, all studies, regardless of their methodological quality, will be included in the data extraction and synthesis process. The review will, however, present and discuss quality issues.

**Data extraction.**   The standardised data extraction tool in JBI SUMARI will be used to extract data from the included studies [31]. The data extracted will include specific details about the population, context, geographical location, study methods, and phenomena of interest relevant to the review objectives, such as experiences of cognitive stimulation by individuals with dementia and their informal caregivers. The findings, and their illustrations, will be extracted verbatim and assigned a level of credibility.

**Data synthesis.**   JBI SUMARI with the meta-aggregation approach will be used to pool qualitative research findings when possible [31]. This will involve the aggregation or synthesis of findings to generate a set of statements that represent the aggregation by assembling and categorising findings according to their similarity in meaning. Following this, these categories will be synthesised to produce a single comprehensive set of synthesised findings that can serve as a basis for evidence-based practice. The two independent reviewers (SR and MC) will initially review all the findings and perform synthesis or aggregation of findings based on similarities in their meaning. Where disagreements may arise, these will be resolved through discussion or involving an additional reviewer (OB). Where textual pooling is not possible, the findings will be presented in narrative format by the same two independent reviewers and confirmed by the review team. Each extracted finding and its accompanying illustrations will be rated by two independent reviewers (SR and MC) based on their perceptions of how well each illustration supports the specific finding associated with it by utilising three levels of credibility in accordance with the JBI methodology for systematic reviews of qualitative data. The three levels of credibility are unequivocal, credible and not supported; where "unequivocal" refers to a finding that is clearly related, supported, and accompanied by illustrations beyond a reasonable doubt, "credible" refers to a finding accompanied by an illustration but the illustration only vaguely demonstrates an association with the finding and "not supported" refers to a finding with no illustration to support it or the provided illustrations are entirely unrelated to the finding. Only unequivocal and credible findings will be included in the synthesis. Data will be disaggregated for people with dementia and informal caregivers where possible. However, since disaggregation of data significantly depends on the availability of individual articles on the topic area and how they analyse and report their findings, such a decision would be made as part of meta-aggregation in accordance with the JBI methodology for systematic reviews of qualitative data. Again, the review findings will be presented per the guidelines provided in the

JBI Manual for Evidence Synthesis. There will be a meta-aggregative schematic/overview flow chart and a meta-aggregative schematic table with relevant narratives to explain the categories and synthesised findings.

**Assessing confidence in the findings.** The final synthesised findings will be graded according to the ConQual approach for establishing confidence in the results of qualitative research synthesis, and will be presented in a Summary of Findings. The Summary of Findings will include the major elements of the review and detail how the ConQual score was developed, the title, population, phenomena of interest and the context for the specific review. The synthesised findings will then be presented along with the type of research that supported them, the score for dependability and credibility, and the overall ConQual score.

**Reflexivity.** It is important to consider the reviewers' assumptions and preconceptions regarding the phenomenon of interest in qualitative systematic reviews, as well as the possible influence these assumptions may have on the review process. The reviewers will therefore make conscious efforts to minimise their preconceived ideas and assumptions from influencing the review by writing memos and reflecting upon these during each step of data collection, analysis and writing the review. During the memo-writing process, methodological notes explaining procedural decisions and observational comments will be written to describe and explore the reviewer's feelings at different stages.

## 3. Discussion

This qualitative systematic review will identify and synthesise the current literature concerning the experience of individuals with dementia who have participated in a cognitive stimulation program, and the experience of their informal caregivers. Evaluating the experience of complex interventions like cognitive stimulation is important in determining their effectiveness, with the caregiver's perspective providing a valuable insight into the real-life impact of these interventions [13]. A variety of approaches to cognitive stimulation interventions exist, with previous reviews summarising the experiences of specific cognitive stimulation programs, like CST [18], or summarising the experience of cognitive stimulation interventions as part of a broader review of non-pharmacological interventions for individuals with progressive memory disorders [20]. This review aims to comprehensively summarise the experience of attending all types of cognitive stimulation programs, identifying perceived benefits, challenges, barriers, and facilitators to this approach to intervention. This evidence will thus be used to inform the implementation and future development of cognitive stimulation interventions for individuals with dementia.

The National Institute for Health and Care Excellence (NICE) guidelines for the assessment, management and support of people living with dementia [33] recommend CST, a common but specific type of cognitive stimulation intervention, be provided to people with mild-to-moderate dementia. While there are no explicit and formal mentions or recommendations for other types of cognitive stimulation interventions in existing clinical practice guidelines, findings from the proposed review of patients' and their informal caregivers' experiences with all types of cognitive stimulation are likely to offer valuable information that can influence best practice guidelines and recommendations with appropriate suggestions regarding the delivery of these interventions. There are some potential limitations to this study. The exclusion of non-English studies due to the limited translation facilities of the researchers is acknowledged as a limitation, as potentially relevant and valuable studies may be missed. Furthermore, synthesis of qualitative evidence is limited by the potential variation in researchers' interpretation of concepts [34]. However, any further additional

limitations identified throughout the completion of the qualitative systematic review will be acknowledged at publication.

## Supporting information

**S1 Checklist. PRISMA-P (Preferred Reporting Items for Systematic review and Meta-Analysis Protocols) 2015 checklist: Recommended items to address in a systematic review protocol.**
(DOCX)

**S1 File. MEDLINE (Ovid) search strategy.**
(DOCX)

## Author Contributions

**Conceptualization:** Simone M. Ryan, Manigandan Chockalingam, Orla Brady.

**Writing – original draft:** Simone M. Ryan, Manigandan Chockalingam.

**Writing – review & editing:** Simone M. Ryan, Manigandan Chockalingam.

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
