## [Decision Letter · Decision Letter 0]

7 May 2023

PONE-D-23-01290

Experiences of patients and their informal caregivers with cognitive stimulation programs for dementia: a qualitative systematic review protocol

PLOS ONE

Dear Dr. Ryan,

Thank you for submitting your manuscript to PLOS ONE. After careful consideration, we feel that it has merit but does not fully meet PLOS ONE’s publication criteria as it currently stands. Therefore, we invite you to submit a revised version of the manuscript that addresses the points raised during the review process.

Minor revisions are required in order for this manuscript to be processed. 

We look forward to receiving your revised manuscript.

Kind regards,

Michael Gilbert McCaul, MSc, PhD

Academic Editor

PLOS ONE

Journal Requirements:

Reviewers' comments:

Reviewer's Responses to Questions

**Comments to the Author**

1. Does the manuscript provide a valid rationale for the proposed study, with clearly identified and justified research questions?

Reviewer #1: Yes

Reviewer #2: Yes

2. Is the protocol technically sound and planned in a manner that will lead to a meaningful outcome and allow testing the stated hypotheses?

Reviewer #1: Yes

Reviewer #2: Partly

3. Is the methodology feasible and described in sufficient detail to allow the work to be replicable?

Reviewer #1: Yes

Reviewer #2: Yes

4. Have the authors described where all data underlying the findings will be made available when the study is complete?

Reviewer #1: Yes

Reviewer #2: Yes

5. Is the manuscript presented in an intelligible fashion and written in standard English?

Reviewer #1: Yes

Reviewer #2: Yes

6. Review Comments to the Author

You may also provide optional suggestions and comments to authors that they might find helpful in planning their study.

Reviewer #1: Thank you for the opportunity to review your manuscript reporting on a systematic review protocol to determine the experiences of individuals with dementia and their caregivers of a cognitive stimulation program. This is an important topic and you have presented a thorough protocol for conducting the review of qualitative studies.

I have some minor suggestions which may improve the quality of the protocol and review.

1. I suggest referring to 'caregiver(s)' as 'informal caregiver(s)' throughout the protocol (including the title). While this clarification is provided in the 'inclusion criteria' section of the manuscript, I believe it would be easier for the reader if you clarified this earlier in the manuscript.

2. Do you plan to conduct any reflexivity exercises with the team who will be synthesising the data?

3. How do you plan to synthesise findings from patients and caregivers? Where possible, will you plan on separating the findings from these two participant groups? Obviously this will be dependent on how individual articles have analysed and reported their findings, but I believe it would be worthwhile outlining how you plan to analyse findings from the two participant groups.

4. I think the discussion could be expanded on by reference to national/international dementia care guidelines and how your findings will complement these guidelines. In recent years there has been an increased focus on dementia care, with an associated increase in the number of clinical practice guidelines, and it would be interesting to consider how your review may contribute to our understanding of current best practice.

Reviewer #2: In review of the protocol ‘Experiences of patients and their informal caregivers with cognitive stimulation programs for dementia: a qualitative systematic review protocol’, I have recommended minor revisions. The proposed qualitative evidence synthesis (QES) will make a strong contribution to the literature. Herewith my minor suggestions for consideration.

Abstract:

16-21: In the first two lines of the abstract, the author tends to foreground the caregivers. I suggest that the voices of those with dementia and the problem statement are highlighted. The rationale for the inclusion of caregivers is not needed in the abstract.

Introduction:

124-134: The description of existing reviews on the similar topic needs to be edited in improve readability. The first review is reported as broad and the second as more specific but neither as comprehensive. Perhaps each review can be critically reported and how the proposed review fills an identified gap.

Methods:

259: Although the methods are well designed, the data synthesis for the meta-aggregation needs some refinement. Specifically, authors should consider providing information on the following.

Who will conduct the synthesis in JBI SUMARI?

Where pooling is not possible who will write the narrative synthesis?

How will credible findings be determined?

Will data be disaggregated for people with dementia and caregivers?

How will the data be presented?

Discussion:

286: The review aim is more descriptive in the discussion that in the abstract in line 19 and in the introduction in line 137. These should be aligned.

Thank you for the opportunity to review this work and I look forward to seeing the findings of the review.

7. PLOS authors have the option to publish the peer review history of their article (what does this mean?). If published, this will include your full peer review and any attached files.

Reviewer #1: No

Reviewer #2: **Yes: **Lynn Hendricks

Please see attached comments

---

## [Author Response · Author response to Decision Letter 0]

11 May 2023

We are pleased to submit our revised qualitative systematic review protocol titled, "Experiences of patients and their informal caregivers with cognitive stimulation programs for dementia: a qualitative systematic review protocol" to your esteemed journal.

Many thanks for the constructive feedback from the reviewers. The manuscript has been revised significantly in response to all the comments. In order to assist the review process, a detailed response to each statement/comment made by the reviewers has been uploaded. We believe that the revised manuscript meets the requirements for publication.

---

## [Editor Report · Decision Letter 1]

13 Jun 2023

Experiences of patients and their informal caregivers with cognitive stimulation programs for dementia: a qualitative systematic review protocol

PONE-D-23-01290R1

Dear Dr. Ryan,

We’re pleased to inform you that your manuscript has been judged scientifically suitable for publication and will be formally accepted for publication once it meets all outstanding technical requirements.

Kind regards,

Michael Gilbert McCaul, MSc, PhD

Academic Editor

PLOS ONE
---

## [Editor Report · Acceptance letter]

19 Jun 2023

PONE-D-23-01290R1 

‘Experiences of patients and their informal caregivers with cognitive stimulation programs for dementia: a qualitative systematic review protocol’ 

Dear Dr. Ryan:

I'm pleased to inform you that your manuscript has been deemed suitable for publication in PLOS ONE. Congratulations! Your manuscript is now with our production department. 

Kind regards, 

on behalf of

Dr. Michael Gilbert McCaul 

Academic Editor

PLOS ONE